

# Ciprofloxacin reduces tenocyte viability and proteoglycan synthesis in short-term explant cultures of equine tendon

Stuart James[1], Johannes Schuijers[1], John Daffy[2], Jill Cook[3] and Tom Samiric[1]

[1] Department of Physiology, Anatomy, and Microbiology, La Trobe University, Melbourne, Victoria, Australia
[2] Department of Infectious Diseases, St. Vincent's Hospital, Melbourne, Victoria, Australia
[3] Sports and Exercise Medicine Research Centre, La Trobe University, Melbourne, Victoria, Australia

Corresponding authors
Stuart James,
stuart.james@latrobe.edu.au
Tom Samiric,
t.samiric@latrobe.edu.au

## ABSTRACT

Fluoroquinolones are an effective, broad-spectrum antibiotic used to treat an array of bacterial infections. However, they are associated with an increased risk of tendinopathy and tendon rupture even after discontinuation of treatment. This condition is known as fluoroquinolone-associated tendinopathy, the underlying mechanisms of which are poorly understood. While many factors may be involved in the pathophysiology of tendinopathies in general, changes in tenocyte metabolism and viability, as well as alteration of proteoglycan metabolism are prominent findings in the scientific literature. This study investigated the effects of ciprofloxacin, a common fluoroquinolone, on cell viability, proteoglycan synthesis, and proteoglycan $m$RNA expression in equine superficial digital flexor tendon explants after 96 h treatment with between 1–300 μg/mL ciprofloxacin, and again after 8 days discontinuation of treatment. Ciprofloxacin caused significant reductions in cell viability by between 25–33% at all dosages except 10 μg/mL, and viability decreased further after 8 days discontinuation of treatment. Proteoglycan synthesis significantly decreased by approximately 50% in explants treated with 100 μg/mL and 300 μg/mL, however this effect reversed after 8 days in the absence of treatment. No significant $m$RNA expression changes were observed after the treatment period with the exception of versican which was down-regulated at the highest concentration of ciprofloxacin. After the recovery period, aggrecan, biglycan and versican genes were all significantly downregulated in explants initially treated with 1–100 μg/mL. Results from this study corroborate previously reported findings of reduced cell viability and proteoglycan synthesis in a whole tissue explant model and provide further insight into the mechanisms underlying fluoroquinolone-associated tendinopathy and rupture. This study further demonstrates that certain ciprofloxacin induced cellular changes are not rapidly reversed upon cessation of treatment which is a novel finding in the literature.

## INTRODUCTION

Fluoroquinolones (FQs) are a group of broad-spectrum antibiotics which have been used for several decades to treat a variety of bacterial infections. While considered relatively safe, there is growing literature indicating an association of FQ use and tendon pathology, particularly of the Achilles tendon (*Ribard et al., 1992*). Although the reported incidence of FQ-induced tendinopathy is low among the general population (<1% of patient prescriptions) (*Gillet et al., 1995*; *Harrell, 1999*; *Wilton, Pearce & Mann, 1996*), in 2008 the US Food and Drug Administration included a black box warning label on almost all prescribed FQs.

FQ treatment has been shown to increase the risk of Achilles tendinopathy ~4-fold and tendon rupture ~2.5-fold; this risk is further increased with concurrent corticosteroid therapy (*Alves, Mendes & Marques, 2019*) and with aging (tendon rupture being ~3× higher for individuals over 60 years of age; *Corrao et al., 2006*). Researchers have reported that pathology could be present as early as 2 h after initial treatment to as late as 6 months after stopping treatment (*Corrao et al., 2006*; *Fernandez-Cuadros et al., 2019*; *Lewis & Cook, 2014*).

The specific mechanism responsible for FQ-induced tendinopathy remains unclear. Previous studies have suggested that the tendon cells (tenocytes) are the primary site of FQ toxicity, including inhibition of tenocyte proliferation and migration (*Burkhardt et al., 1990*; *Williams et al., 2000*), reduced mitochondrial activity (*Bernard-Beaubois et al., 1998*), changes in expression of inflammatory mediators, and reduced collagen and proteoglycan synthesis (*Bernard-Beaubois et al., 1998*; *Williams et al., 2000*). Other authors have shown that FQs interfere directly with the extracellular matrix proteins (*Bendele et al., 1990*).

Tenocytes are modified fibroblasts which are responsible for the synthesis, maintenance, and degradation of the extracellular matrix. This matrix is comprised predominantly of parallel-arranged Type I collagen fibres with proteoglycans and other non-collagenous proteins interspersed. Although proteoglycans make up <3% of the tendon matrix, these macromolecules contribute significantly to the structural integrity of the tissue.

Proteoglycans are a heavily glycosylated protein consisting of a core protein with one or more sulphated glycosaminoglycan (sGAG) chains covalently attached. The sGAGs typically act to sequester cations and subsequently water, providing hydration, stability, and pressure to the tissue to improve compressive strength. We have previously shown that the small leucine-rich proteoglycans, decorin and biglycan, make up approximately 80% of total proteoglycans in energy-storage tendons such as the Achilles (*Samiric, Ilic & Handley, 2004*). These are predominantly involved in regulating collagen fibrillogenesis; in their absence, collagen fibrils are coarse, irregular and disorganized (*Danielson et al., 1997*). The large aggregating proteoglycans, aggrecan and versican, make up the remainder and due to their abundance of sGAGs, provide the tissue with a high capacity to resist compressive forces associated with loading and mobilisation. Increased levels of aggrecan have been associated with tendon pathology, causing increased hydration and swelling of the tissue (*Riley, 2008*).

The aim of this study was to utilise an explant culture system using tendons derived from equine superficial digital flexor tendon (SDFT) to examine the effects of ciprofloxacin (CPX) on tendon proteoglycan synthesis. This study further aimed to determine whether changes in tenocyte viability or proteoglycan synthesis are associated with discontinuation of the antibiotic.

## MATERIALS & METHODS

### Materials used

Ciprofloxacin lactate solution (100 mg/50 mL) was obtained from Sandoz (NSW, Australia) for use in tissue explant culture. Dulbecco's Modified Eagle's Medium (DMEM), penicillin-streptomycin, newborn calf serum (NBCS) and PrestoBlue® Reagent were purchased from Life Technologies (NY, USA). Sulfur-35 radionuclide and ScintiSafe 30% was purchased from PerkinElmer (Boston, MA, USA). PD-10 (Sephadex G-25) columns were obtained from GE Life Sciences (Uppsala, Sweden). Papain and Sepharose CL-4B resin were purchased from Sigma–Aldrich (St. Louis, MO, USA). RNAlater was from Qiagen (Hilden, Germany). PureZOL reagent, Fatty and fibrous tissue RNA isolation kits (Cat#732-6820), RNA-cDNA reverse transcription kits (Cat#170-8890), and an iCycler IQ detection system were purchased from Bio-Rad (Hercules, CA, USA). Bertin Precellys 24 homogeniser and CK 28 ceramic homogeniser beads were purchased from Bertin Technologies (France). NanoDrop 2000 was bought from Thermo Fisher Scientific (Waltham, MA, USA). Premier Biosoft (Palo Alto, CA, USA) developed beacon primer design software. GraphPad (La Jolla, CA, USA) provided prism 5.0 data analysis software. No ethical approval was required for this study as animals were not euthanized for the purposes of this study and all tissue was kindly provided by Tooradin Knackery, Tooradin, Victoria, Australia.

### Tissue explant culture

The mid-carpal segments (~5 cm) of the superficial digital flexor tendon (SDFT) were harvested under aseptic conditions from the forelimbs of Thoroughbred horses euthanized for reasons unrelated to tendon disease. Specimens from up to eight individual adult equine (~6 years) were obtained from a local knackery.

The paratenon was removed and the remaining samples of core tendon were cut into small pieces weighing ~100 mg each. These were incubated for 24 h at 37 °C in low glucose DMEM (10 mL per 1 g tissue) supplemented with 10% (v/v) NBCS and 1% (v/v) penicillin/streptomycin. Duplicate samples of approximately 100 ± 20 mg tissue were then distributed between treatment groups and placed into individual sterile screw-capped vials containing the same DMEM supplemented with 10% NBCS, as described above, alone (control) or containing ciprofloxacin (treatment).

### Ciprofloxacin treatment

CPX-treated and control (CT) tissue explants were cultured in DMEM with 10% NBCS for 96 h using duplicate cultures for each sample. CPX was added to the culture medium of experimental groups at four different doses: 1, 10, 100 and 300 μg/mL. The lower range of

concentrations was selected to include dosages that reflect serum CPX concentrations reported clinically, and higher dosages are consistent with similar *in vitro* toxicological studies. CT samples were those in DMEM alone. The culture medium was changed after 48 h.

In parallel experiments investigating whether discontinuation of CPX affects tendon metabolism, at the end of the 96 h CPX treatment period, samples were thoroughly washed using sterile PBS and placed in DMEM alone. These samples were incubated for a further 8 days under control conditions, with media replenished every 48 h.

## PrestoBlue® reagent assay

The effect of CPX on total cell viability in explant cultures of equine tendon after 96 h of treatment and subsequent 8-day recovery period was measured by quantification using a PrestoBlue® reagent assay. The tendon explants were incubated with DMEM containing 10% (v/v) PrestoBlue® reagent for 3 h at 37 °C. Following this, 200 μL reaction mixtures were transferred to 96-well plates for measurement of fluorescence (Triad MX multimeter reader) with an excitation wavelength of 530 nm and an emission wavelength of 590 nm. DMEM with 10% PrestoBlue® reagent, without any tendon explant, was used as a reagent blank for the fluorometric measurements. The tendon explants were rinsed thoroughly in sterile PBS to remove any residual PrestoBlue® reagent after each assay.

## Determination of sulphated glycosaminoglycan concentration

A dimethylmethylene blue (DMB) spectrophotometric assay (*Farndale, Buttle & Barrett, 1986*) was used to quantify the level of sulphated glycosaminoglycans (sGAG) in papain-digested tendon samples. Approximately 10 mg wet weight tissue was lyophilised to determine tissue dry weight and then digested with 125 μg/mL papain, 10 mM cysteine hydrochloride, 2 mM EDTA and 0.1 M phosphate buffer (0.2 M monosodium dihydrogen orthophosphate and 0.2 M disodium hydrogen orthophosphate; pH 7.0) for 18 h at 65 °C. Triplicate aliquots of the papain-digested tendon samples were immediately analysed at an absorbance 525 nm (*Farndale, Buttle & Barrett, 1986*). The assay was calibrated by use of standards containing up to 40 μg/mL bovine chondroitin sulphate. The sGAG concentration in the tendon samples was obtained by comparison with the standard curve and expressed as μg of chondroitin sulphate per mg dry weight of tissue.

## Determination of newly synthesised [35]S-labelled proteoglycans

The amount of newly synthesised proteoglycans by tendon explants was measured by [35]S-sulphate labelling. At the end of the culture periods, tissue samples were washed thoroughly in sulphate-free medium and then placed in sulphate-free medium containing 150 μCi/mL [35]S-sulphate at 37 °C for 6 h. A single batch of medium containing [35]S-sulphate was used for all incubations. At the end of the incubation period, [35]S-labelled proteoglycans were extracted with 4 M Guanidine Hydrochloride (GnHCl; pH 6.1) in the presence of a proteinase inhibitor cocktail at 4 °C for 72 h, followed by 0.5 M NaOH at room temperature for 24 h.

Aliquots (0.5 mL) of the tissue extracts were applied to PD-10 (Sephadex G-25) columns, equilibrated and eluted with 4M GnHCl, 0.1 M sodium sulphate, 0.05 M sodium acetate, 0.1% (v/v) Triton X-100; pH 6.1. Aliquots of the eluted samples were then collected
**Table 1** Oligonucleotide sequences for the specific equine primers used in RT-PCR.

| Gene | Accession number | Base pair length | Sense | Antisense |
|------|------------------|------------------|-------|-----------|
| GAPDH | DQ403057 | 138 bp | GGCACCGTCAAGGCTGAGAAC | GGTGAAGACGCCAGTGGACTC |
| Aggrecan | NM_001135 | 127 bp | TGCGTGGGTGACAAGGACAG | CAAGGCGTGTGGCGAAGAAC |
| Versican | NM_004385 | 147 bp | ATCTGGATGGTGATGTGTTC | AATCGCAACTGGTCAAAGC |
| Decorin | BC005322 | 143 bp | CTGGGCTGGACCGTTTCAAC | GATGGCATTGACAGCGGAAGG |
| Biglycan | BC002416 | 139 bp | ACACCATCAACCGCCAGAGTC | GACAGCCACCGACCTCAGAAG |
| Fibromodulin | BC035281 | 145 bp | GGCTGCTCTGGATTGCTCTC | CGGGTCAGGTTGTTGTGGTC |

in vials containing a scintillant cocktail (ScintiSafe 30%), and the radioactivity in each sample assessed with a Packard 1500 TRI-CARB liquid scintillation counter.

## RNA extraction and reverse transcriptase

At the end of the culture period tissue samples (~50–100 mg wet weight) were placed in RNAlater™ at 4 °C overnight, and then stored at −80 °C for future analysis. Tissue was homogenised in PureZOL reagent using a Bertin Precellys 24 homogeniser with the aid of ceramic beads (CK 28) and a program consisting of 6,000$g$ for 20 s. This was repeated five times until all visible tissue was dissolved. Following homogenisation, total RNA was isolated using a kit according to the recommendations of the manufacturer. The procedure included digestion of genomic DNA with DNase I. RNA integrity and concentration were determined using a Nanodrop 2000. Complementary DNA (cDNA) was synthesised using a reverse transcription kit. The resulting cDNA was subjected to real-time PCR amplification in an iCycler iQ Detection System (Bio-Rad, Hercules, CA, USA). Equine oligonucleotide sequences of the specific primers used in this study were designed using Beacon Designer and are shown in Table 1.

## Quantification of gene expression

The values obtained for $m$RNA expression for the genes of interest were normalised for GAPDH housekeeping gene in the same sample. This was calculated according to relative quantification using the ΔΔCt method where Ct is the cycle number of the detection threshold, and ΔΔCt shows the difference in threshold cycle (ΔCt) between gene of interest and GAPDH.

## Data analysis

Results are represented as mean ± standard error of the mean (SEM). Experiments were repeated on separate occasions using tendon tissue derived from a different horse each time. Each datum from an individual experiment is compared to a CT sample from the same animal. Overall experiment data are presented as a mean of all tissue samples compared to their respective control after 96 h treatment or 96 h treatment followed by 8 days in CT conditions (recovery). All statistical tests were performed using Prism 9.0 software. Repeated measures one-way analysis of variance (ANOVA) with Greenhouse-Geisser correction followed by post-hoc Dunnett's-test was used to determine multiplicity
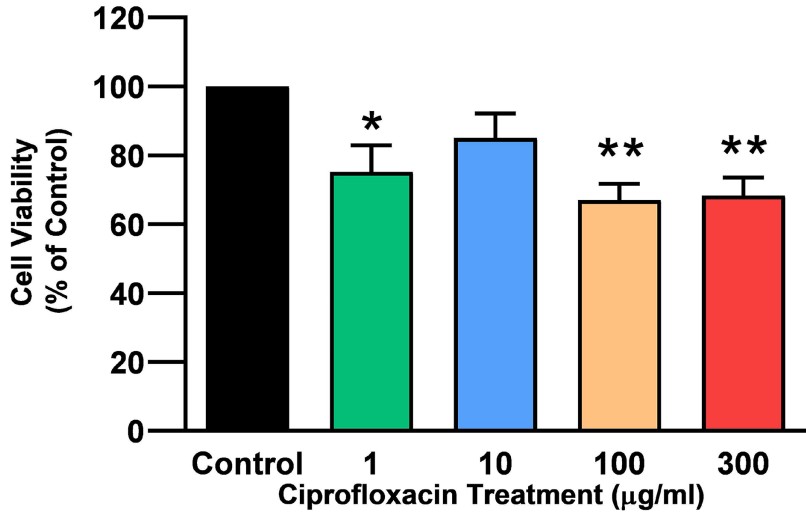

**Figure 1 Cell viability, as determined by PrestoBlue® reagent assay, in equine derived tendon explant cultures after 96 h treatment with 1, 10, 100 & 300 μg/mL CPX.** Values represent mean ± SEM, expressed as % of control (CT). *$p < 0.01$, **$p < 0.001$ compared with control. $n = 4$.

adjusted significance between each treatment condition and control. Differences were considered statistically significant at $p < 0.05$.

# RESULTS

## Effect on tenocyte viability

Tendon explant cultures established from SDFTs from 6-year old horses were exposed to 1–300 μg/mL CPX for 96 h. The PrestoBlue® viability assay revealed that 1, 100 and 300 μg/mL CPX reduced cell viability by approximately 25% ($p = 0.02$), 33% ($p = 0.003$), and 32% ($p = 0.004$) compared to CT, respectively (Fig. 1). Although, a downward trend was observed in samples exposed to 10 μg/mL CPX, this was not shown to be significant ($p > 0.05$). Eight days after discontinuation of treatment, tenocyte viability was reduced in a dose dependent manner (Fig. 2) by approximately 30% ($p = 0.048$), 34% ($p = 0.043$), 36% ($p = 0.02$), and 51% ($p = 0.0036$) compared to CT in samples initially treated with 1, 10, 100, & 300 μg/mL CPX, respectively.

## Effects on sulphated glycosaminoglycan levels (sGAG)

The sGAG levels present in the tissue following treatment with 1–300 μg/mL CPX for 96 h was determined by a DMB spectrophotometric assay. There was no significant change in sGAG levels in CPX-treated tendon explants compared with CT ($p > 0.05$; Fig. 3).

## Effects on newly synthesised [35]S-labelled proteoglycan concentrations

The rate of incorporation of radiolabeled sulphate into newly synthesised PGs following treatment with CPX is shown in Fig. 4. The amount of [35]S-sulphate incorporation into PGs was reduced by approximately 48% ($p = 0.0037$) and 51% ($p = 0.0002$) in explant

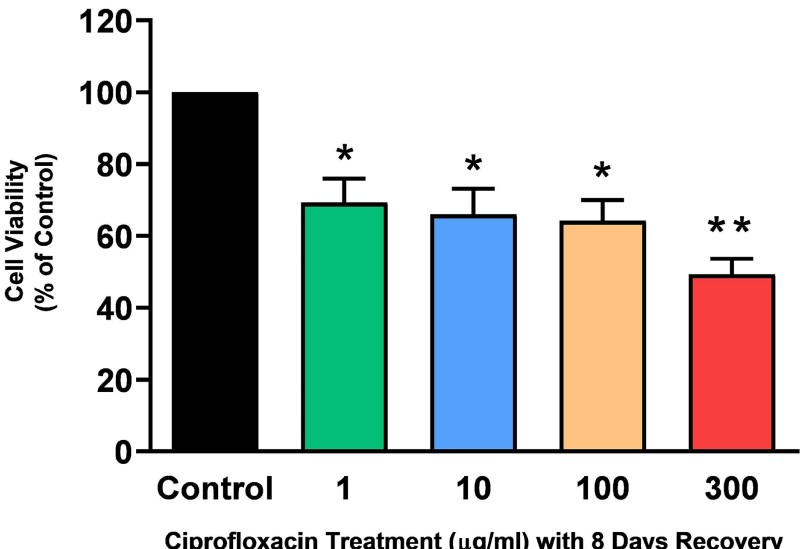

**Figure 2 Cell viability, as determined by PrestoBlue® reagent assay, in equine derived tendon explant cultures after 96 h treatment with 1, 10, 100 & 300 µg/mL CPX and subsequent 8-day recovery period.** Values represent mean ± SEM, expressed as % of control (CT). *$p < 0.05$, **$p < 0.01$ compared with control. $n = 7$.

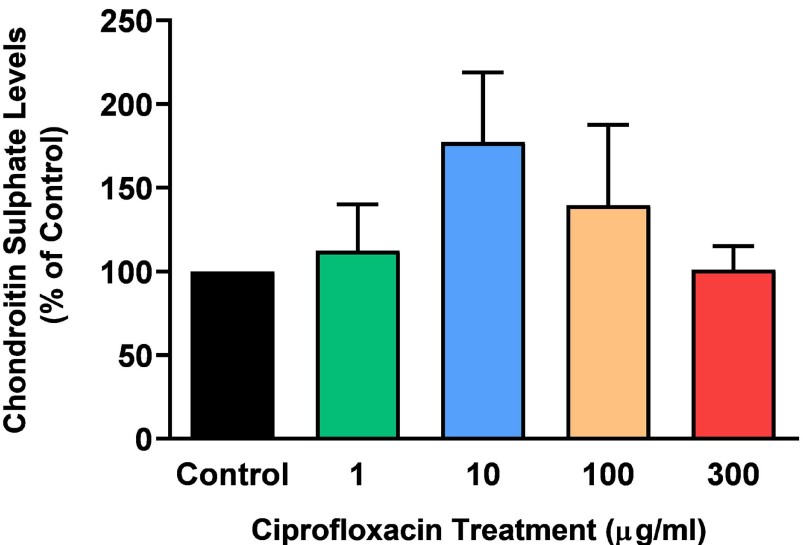

**Figure 3 Chondroitin sulphate levels in equine derived tendon explants cultures after 96 h treatment with 1, 10, 100 or 300 µg/mL CPX.** Values represent mean ± SEM, expressed as % of control (CT). $n = 3$.

cultures treated with 100 and 300 µg/mL CPX, respectively. $^{35}$S-sulphate incorporation into PGs in samples treated with 1, 100, and 300 µg/mL was at 101%, 115%, and 85% of CT levels 8 days after discontinuation of CPX, respectively ($p > 0.05$). $^{35}$S-sulphate incorporation into PGs was increased by 40% in samples treated with 10 µg/mL CPX after the recovery period ($p = 0.0147$; Fig. 5).

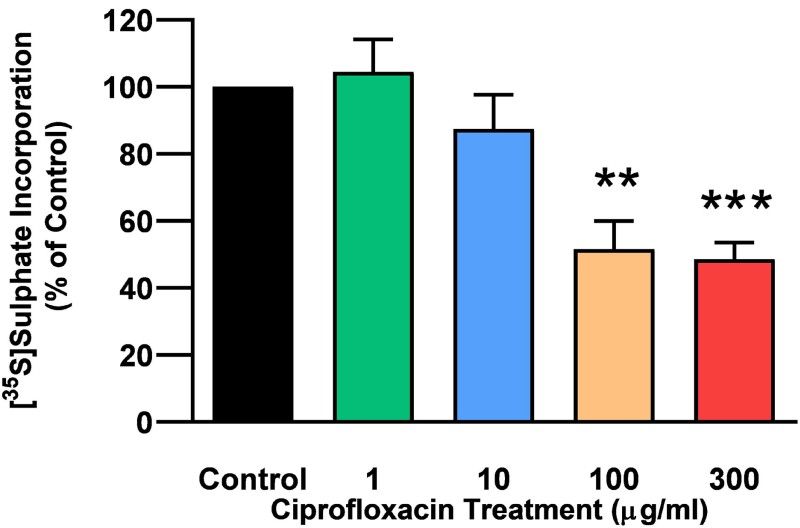

**Figure 4** $^{35}$S-sulphate incorporation in equine derived tendon explants cultures after 96 h treatment with 1, 10, 100 or 300 µg/mL CPX. Values represent mean ± SEM, expressed as % of control (CT). $^{**}p < 0.01$, $^{***}p < 0.001$ compared with control. $n = 6$.

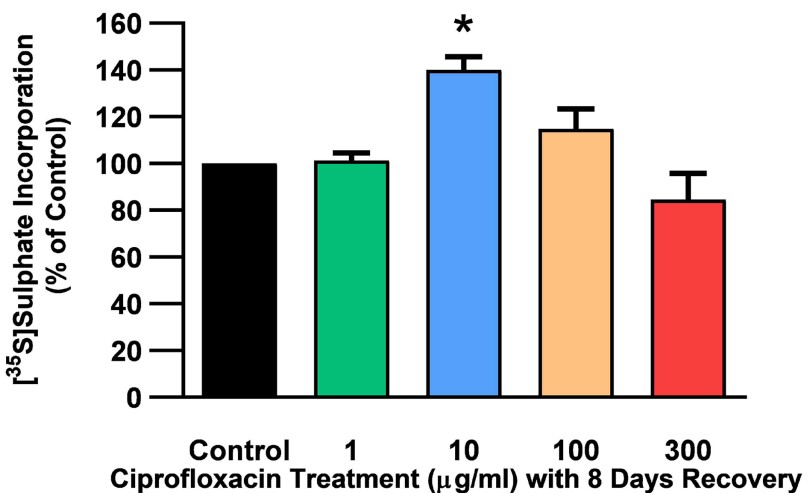

**Figure 5** $^{35}$S-sulphate incorporation in equine derived tendon explants cultures after 96 h treatment with 1, 10, 100 or 300 µg/mL CPX and subsequent 8-day recovery period. Values represent mean ± SEM, expressed as % of control (CT). $^{*}p < 0.05$ compared with control. $n = 4$.

### Effects on mRNA expression of proteoglycan genes

We analysed several PG genes including decorin, fibromodulin, biglycan, aggrecan and versican in CPX-treated and untreated explants (Figs. 6–10). The only significant change in *m*RNA expression of any PG gene observed after 96 h treatment with CPX was a 0.49-fold down-regulation of versican at 300 µg/mL ($p = 0.001$; Fig. 10). Eight days after discontinuation of CPX treatment, no significant differences were observed in *m*RNA expression of decorin or fibromodulin (Figs. 11 and 12). However, biglycan *m*RNA was

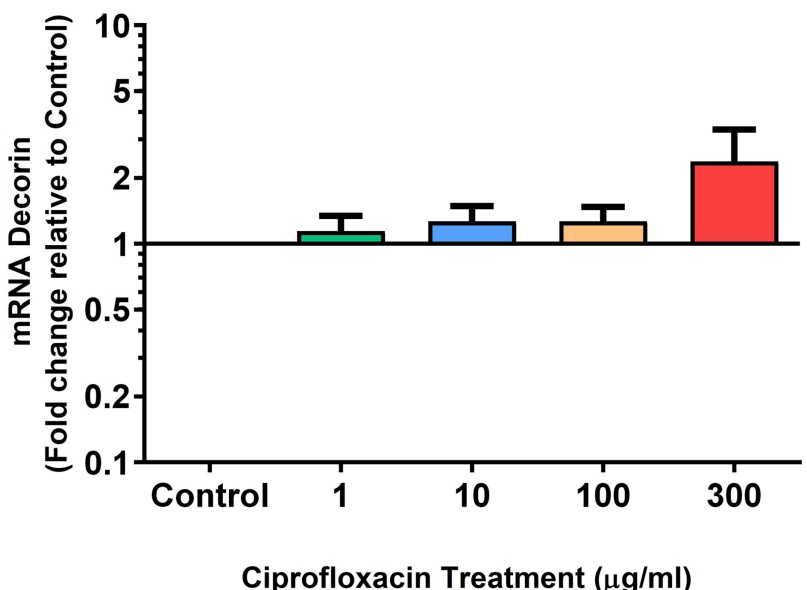

**Figure 6** *m*RNA expression of decorin in equine derived tendon explant cultures after 96 h treatment with 1, 10, 100 & 300 μg/mL CPX. Results are expressed logarithmically. Values represent mean ± SEM, expressed as fold change relative to control, normalised to GAPDH. $n = 6$.

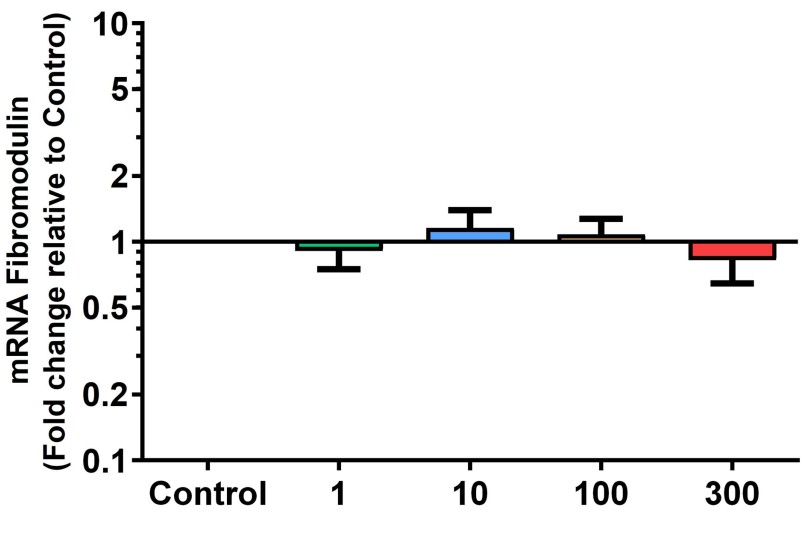

**Figure 7** *m*RNA expression of fibromodulin in equine derived tendon explant cultures after 96 h treatment with 1, 10, 100 & 300 μg/mL CPX. Results are expressed logarithmically. Values represent mean ± SEM, expressed as fold change relative to control, normalised to GAPDH. $n = 6$.

down-regulated to 0.64-fold ($p = 0.001$), and 0.67-fold ($p = 0.0095$) of CT in explants originally treated with 10, and 100 μg/mL CPX (Fig. 13). Aggrecan was down-regulated to 0.56-fold ($p = 0.007$) and 0.45-fold ($p = 0.01$) of CT after 8 days recovery in samples treated

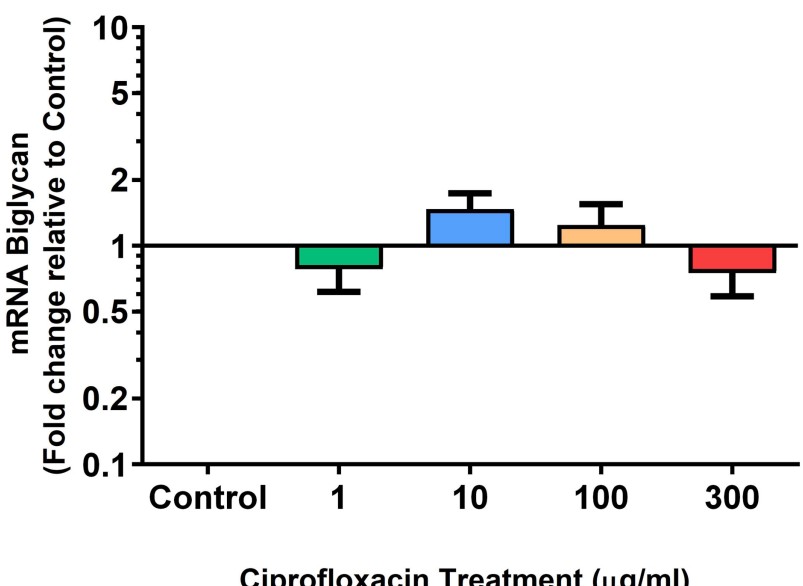

**Figure 8 *m*RNA expression of biglycan in equine derived tendon explant cultures after 96 h treatment with 1, 10, 100 & 300 µg/mL CPX.** Results are expressed logarithmically. Values represent mean ± SEM, expressed as fold change relative to control, normalised to GAPDH. *n* = 6.

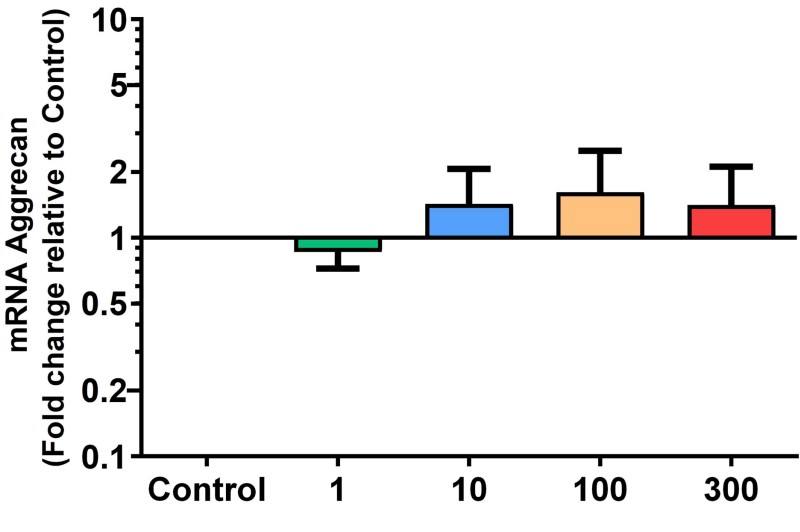

**Figure 9 *m*RNA expression of aggrecan in equine derived tendon explant cultures after 96 h treatment with 1, 10, 100 & 300 µg/mL CPX.** Results are expressed logarithmically. Values represent mean ± SEM, expressed as fold change relative to control, normalised to GAPDH. *n* = 6.

with 1, and 10 µg/mL CPX (Fig. 14). Likewise, versican *m*RNA was down-regulated after 8 days discontinuation of treatment by 0.5-fold ($p = 0.0032$) compared to CT in explants initially treated with 1 µg/mL CPX (Fig. 15).

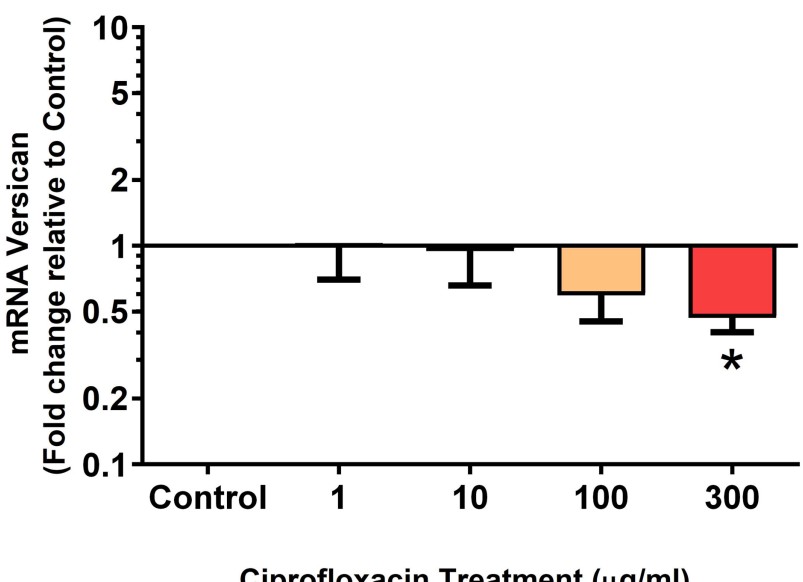

**Figure 10 *m*RNA expression of versican in equine derived tendon explant cultures after 96 h treatment with 1, 10, 100 & 300 μg/mL CPX.** Results are expressed logarithmically. Values represent mean ± SEM, expressed as fold change relative to control, normalised to GAPDH. *$p < 0.05$ compared with control. $n = 6$.  

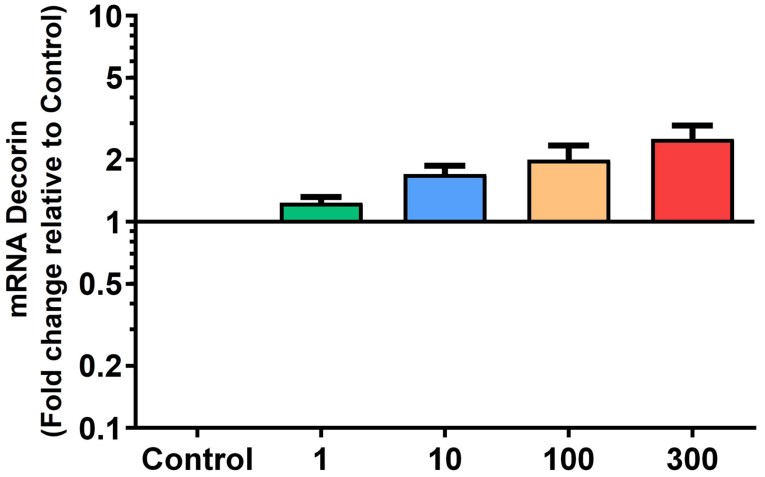

**Figure 11 *m*RNA expression of decorin in equine derived tendon explant cultures after 96 h treatment with 1, 10, 100 & 300 μg/mL CPX followed by 8 days in control conditions.** Results are expressed logarithmically. Values represent mean ± SEM, expressed as fold change relative to control, normalised to GAPDH. $n = 4$.  

## DISCUSSION

Changes in tendon metabolism following FQ treatment in a cell culture system are well established in the literature across multiple species including canine (*Lim et al., 2008*; *Williams et al., 2000*), avian (*Yoon et al., 2004b*), equine (*Yoon et al., 2004a*), rabbit

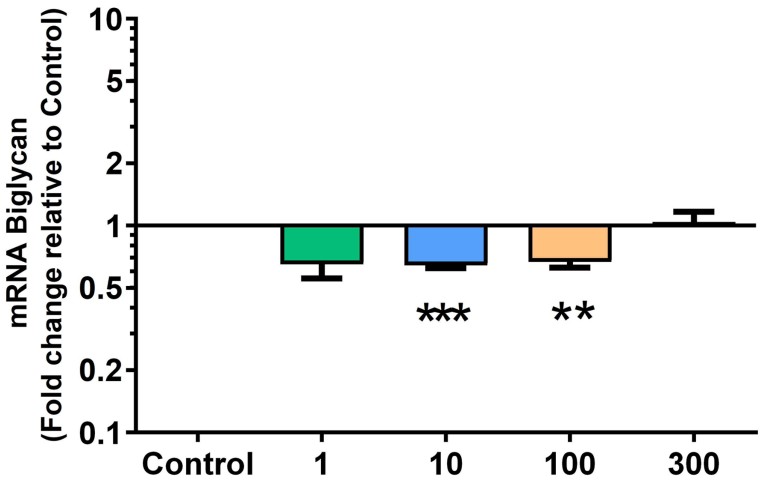

**Figure 12 *m*RNA expression of fibromodulin *m*RNA expression in equine derived tendon explant cultures after 96 h treatment with 1, 10, 100 & 300 μg/mL CPX followed by 8 days in control conditions.** Results are expressed logarithmically. Values represent mean ± SEM, expressed as fold change relative to control, normalised to GAPDH. **$p < 0.01$, ***$p < 0.001$ compared with control. $n = 4$.

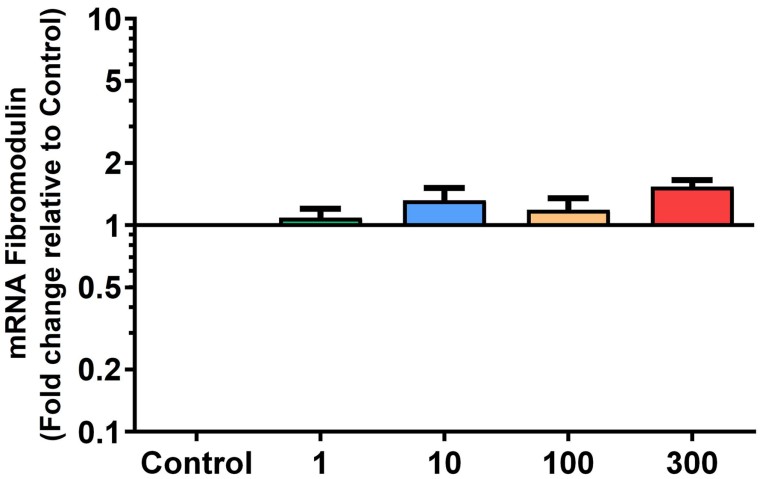

**Figure 13 *m*RNA expression of biglycan in equine derived tendon explant cultures after 96 h treatment with 1, 10, 100 & 300 μg/mL CPX followed by 8 days in control conditions.** Results are expressed logarithmically. Values represent mean ± SEM, expressed as fold change relative to control, normalised to GAPDH. $n = 4$.

(*Pouzaud et al., 2004*) and human (*Lowes et al., 2009*). Our study demonstrated that CPX not only inhibits tenocyte viability but also reduces newly synthesised PGs in equine tendon explant cultures.

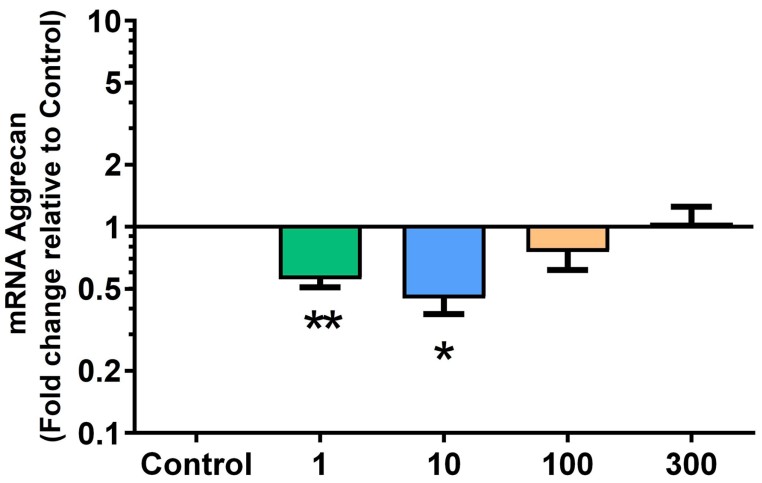

**Figure 14** *m*RNA expression of aggrecan in equine derived tendon explant cultures after 96 h treatment with 1, 10, 100 & 300 µg/mL CPX followed by 8 days in control conditions. Results are expressed logarithmically. Values represent mean ± SEM, expressed as fold change relative to control, normalised to GAPDH. *$p < 0.05$, **$p < 0.01$ compared with control. $n = 4$.

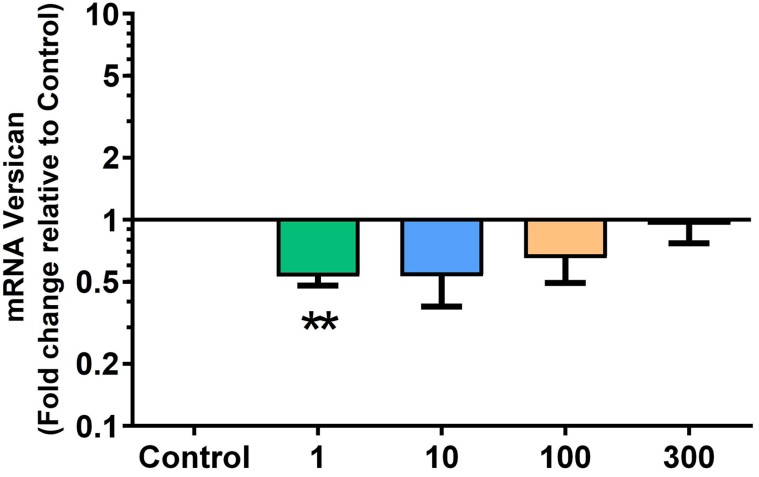

**Figure 15** *m*RNA expression of versican in equine derived tendon explant cultures after 96 h treatment with 1, 10, 100 & 300 µg/mL CPX followed by 8 days in control conditions. Results are expressed logarithmically. Values represent mean ± SEM, expressed as fold change relative to control, normalised to GAPDH. **$p < 0.01$ compared with control. $n = 4$.

To provide a basis for extrapolating these results to clinical practice, we considered the pharmacokinetic properties of FQs in our study (*Stahlmann et al., 1995*; *Zhanel & Noreddin, 2001*; *Zhanel et al., 2001*). The lowest range of concentrations tested

(1–10 µg/mL) reflect plasma concentrations seen in humans treated with CPX. Our highest concentrations used in this study (100–300 µg/mL) are common in such *in vitro* FQ toxicological studies and are consistent with highest concentrations used by other investigators (*Egerbacher, Edinger & Tschulenk, 2001*; *Lim et al., 2008*; *Yoon et al., 2004a*; *Yoon et al., 2004b*). Despite apparent articular tissue accumulation of FQs, with concentrations as high as approximately 30 µg/mL reported in articular cartilage in canines treated with ofloxacin (*Yoshida et al., 1998*), such concentrations have not been demonstrated to occur in tendon tissue *in vivo* and peak concentration of CPX achieved in tendon is unknown (*Lewis & Cook, 2014*). As such, the effects observed in this study at higher dosages may have limited applicability to clinical settings.

The mechanism by which CPX suppressed tenocyte viability is unknown, however other investigators have shown that it may be due to a number of factors including altered metabolism of signalling molecules, such as down-regulation of cytokines involved in G2/M cell cycle arrest and an increase in FAK phosphorylation (*Tsai et al., 2011*), a decrease in ß1-intergin expression and upregulation of caspase-3 (*Sendzik et al., 2005*), as well as changes in epigenetic metabolism (*Badal, Her & Maher, 2015*). Further studies have demonstrated increases in biomarkers of mitochondrial oxidative stress, as well as deletion or double stranded breaks in *mt*DNA preceding loss of viability in a number of mammalian cell cultures as a result of FQ treatment (*Badal, Her & Maher, 2015*; *Lawrence et al., 1996*; *Pouzaud et al., 2004*). Moreover, studies show that decorin has the potential to inhibit cell proliferation *via* TGF-β (*Ruoslahti & Yamaguchi, 1991*; *Ruoslahti et al., 1992*) and *via* the down-regulation of EGF receptors (*Csordas et al., 2000*). Considering our study demonstrated decreased overall PG synthesis induced by CPX, we suggest that suboptimal extracellular concentrations of decorin may be contributing to reduced tenocyte viability in our study. However, the decrease in PG synthesis observed in the current study cannot be solely attributed to one PG and further studies are needed to identify whether decorin metabolism is specifically affected by CPX.

We also showed an inability of tenocyte viability in whole tissue explants to recover following 8 days cessation of CPX treatment. This finding suggests a potential contributing mechanism to the latency between cessation of FQ therapy and development of tendon pathology reported in the literature which can be as high as 6 months post-therapy. Studies have shown that tenocytes are able to monitor and respond to changes in their mechanical environment (*Arnoczky et al., 2004*). Consequently, we speculate that prolonged reductions in tenocyte viability would result in diminished ability of tendons to appropriately monitor the extracellular environment and adapt to load, which is shown to be a feature of early-stage tendinopathy (*Cook et al., 2016*; *Maffulli, Sharma & Luscombe, 2004*).

Measuring [35]S-incorporation into the tissue allows the study of sulphation of GAGs covalently bound to the PG core protein (*Beresford et al., 1987*; *Vogel & Hernandez, 1992*). This approach allowed us to measure the CPX-induced changes on PG synthesis. Our findings corroborate the results of *Williams et al. (2000)* which found that in fibroblasts derived from Achilles tendon and paratenon, CPX (5–50 µg/mL) reduced PG synthesis in a dose dependent manner by up to 53% (*Williams et al., 2000*). Similar

findings were also reported in canine chondrocytes (*Burkhardt et al., 1993*) and chicken tendon-derived cells (*Yoon et al., 2004a*) treated with difloxacin and enrofloxacin, respectively. Furthermore, enrofloxacin has demonstrated an ability to induce changes in the number of *N*-linked oligosaccharides attached to the decorin core protein (*Yoon et al., 2004b*). Our findings also support *in vivo* evidence that showed FQs significantly decreased [35]S-labelled PG synthesis within 24 h in mouse cartilage and Achilles tendon (*Simonin et al., 2000*). Thus, our results demonstrate that short-term exposure to CPX induces cellular changes which lead a decrease in PG synthesis. No significant change in GAG.

Our results also showed that [35]S-labelled PG synthesis in tissue explants recovered 8 days following discontinuation of CPX (concentrations >100 μg/mL, as well as 1 μg/mL). Although the mechanism underlying this recovery is unknown, it is unlikely to be due to a concurrent rebound in tenocyte viability as this was not shown to occur in our study. We speculate that upon removal of CPX, tenocytes may be compensating for the reduction in PG synthesis induced by higher concentrations of CPX, even with reduced tenocyte viability. Indeed, we have previously shown that PGs in tendon are able to be turned over rapidly (within days) when compared to collagen (*Samiric, Ilic & Handley, 2004*). This may also, in part, explain the marked increase in PG synthesis during recovery in tissue explants initially treated with 10 μg/mL. Interestingly, this finding is in contradiction to a study using multiparametric MRI of healthy human Achilles tendons which showed an ~25% reduction in sodium signal (a means of detecting GAG content) 10 days after treatment with CPX (*Juras et al., 2015*). However, these authors also reported that the GAG-related sodium signal returned to baseline at a 5-month follow up, and that none of the subjects involved developed any clinical symptoms of Achilles tendinopathy. This suggests that *in vivo* recovery of PG metabolism after CPX treatment is slower than what we have observed *in vitro*. Whether delayed *in vivo* recovery of PG synthesis also contributes to the latency of symptom onset clinically is yet to be established.

While we showed CPX down-regulated versican *m*RNA, this is unlikely to explain the demonstrated reduction in PG synthesis as this only occurred at the highest dosage. Additionally, the down-regulation in *m*RNA expression of biglycan, aggrecan and versican at various initial CPX concentrations shown after 8 days discontinuation of treatment coincided with an observed return to control level [35]S-labelled PG synthesis. This suggest a mechanism outside *m*RNA expression changes and raises the possibility that the observed reduction in PG synthesis is attributable to direct alteration of glycosylation and/or post-translational regulation of PGs rather than their *m*RNA expression. Indeed, it has been reported previously that total monosaccharide content is reduced in equine tenocytes treated with enrofloxacin (*Yoon et al., 2004a*). However, we were unable to support these findings in our current study as no significant changes in GAG concentration were shown after the treatment period. The variability in these data make speculation around the underlying mechanism of reduced PG synthesis difficult and warrants further investigation.

Proper tenocyte and matrix metabolism are vital for tendon adaptation to load as well as healing after injury or pathology. Thus, we propose that the observed effects in this study induce changes that predispose a tendon to structural pathology during or after load

application, and any subsequent development of structural pathology increases the risk of rupture. This may explain why FQ-associated tendinopathy preferentially affects high load and weight bearing tendons, particularly the Achilles tendon (*Lewis & Cook, 2014*). Under normal circumstances, high strain tendons undergo more rapid matrix turnover than low strain tendons in order to be able to rapidly adapt to increased amounts of load (*Bank et al., 1999*). Affecting tendons that are already limited in their capacity to heal would also explain the increased prevalence of FQ-associated tendinopathy amongst the elderly and those concurrently using corticosteroids (*Khaliq & Zhanel, 2003*).

## CONCLUSIONS

In summary, our results demonstrate that both tenocyte viability and synthesis of PGs are suppressed in explants of equine tendon following treatment with CPX. This corroborates previously published literature which has found reduction of tenocyte viability and PG synthesis *in vitro* in numerous species and confirms that these effects occur in a whole tissue explant culture which more closely simulates the *in vivo* environment on tenocytes. These results suggest that administration of fluoroquinolones such as CPX may alter the ability of tendons to undergo normal adaptive turnover, even after discontinuation of FQ therapy. This study further demonstrates that CPX-induced reduction in tenocyte viability does not recover within 8 days of removal of CPX, and we suggest this may be a contributing mechanism to the delayed onset of pathology reported clinically after FQ therapy.

We suggest future research aim to investigate the effects of fluoroquinolones on PG catabolism, as well as characterization of the metabolism of individual PGs during and after FQ treatment. This would provide a complete picture regarding the effects of FQs on PG metabolism in tendon. And, future investigations should attempt to reproduce the observed effects by us and previous investigators of FQs on tendon PG metabolism in a live animal model.

## ACKNOWLEDGEMENTS

Ciprofloxacin lactate was kindly provided by Dr. John Daffy, Head of Infectious Diseases at St. Vincent's Hospital, Melbourne, Australia.

### Funding

Stuart James received the La Trobe APA/RTP post graduate research scholarship throughout the course of data collection and writing. Laboratory consumables and reagents were purchased using internal research funding provided by La Trobe University, Melbourne, Australia. There was no additional external funding received for this study. The funders had no role in study design, data collection and analysis, decision to publish, or preparation of the manuscript.

## Grant Disclosures

The following grant information was disclosed by the authors:
La Trobe APA/RTP Post Graduate Research Scholarship.
La Trobe University, Melbourne, Australia.

## Competing Interests

The authors declare that they have no competing interests.

## Author Contributions

- Stuart James conceived and designed the experiments, performed the experiments, analyzed the data, prepared figures and/or tables, authored or reviewed drafts of the paper, and approved the final draft.
- Johannes Schuijers conceived and designed the experiments, authored or reviewed drafts of the paper, and approved the final draft.
- John Daffy conceived and designed the experiments, authored or reviewed drafts of the paper, and approved the final draft.
- Jill Cook conceived and designed the experiments, authored or reviewed drafts of the paper, and approved the final draft.
- Tom Samiric conceived and designed the experiments, authored or reviewed drafts of the paper, and approved the final draft.

## Data Availability

   All raw data are available in the Supplemental Files.

## Supplemental Information

Supplemental information for this article can be found online at http://dx.doi.org/10.7717/peerj.12003#supplemental-information.

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
