# Peer review of "Ciprofloxacin reduces tenocyte viability and proteoglycan synthesis in short-term explant cultures of equine tendon"

_PeerJ, doi:10.7717/peerj.12003_

## Round 0.1 · original submission · Major Revisions

Please provide assurance in your manuscript that you either received Institutional Review Board approval for this research, or that such approval was not required for it.

·

Basic reporting

Pls. see document attached

Experimental design

Pls. see document attached

Validity of the findings

Pls. see document attached

Additional comments

Pls. see document attached

Reviewer 2 ·

Basic reporting

I want to congratulate the authors for the manuscript entitled: "Ciprofloxacin reduces tenocyte viability and proteogycan synthesis in short-term explant cultures of equine tendon" reference 58014.
The manuscrpit is well-structured, good background has been provided.
The hypothesis is very relevant, since the results justify the clinical observations of fluoroquinolones on tendoncyte viability/repture.
Although some references are older than 10 years, the articles are referred to some recognized or classical in vitro and in vivo studies on the effect of FQ (fluoroquinolones) on tenocyte viability.

Experimental design

I want to congratulate the authors for the manuscript entitled: "Ciprofloxacin reduces tenocyte viability and proteogycan synthesis in short-term explant cultures of equine tendon" reference 58014.
The design is very interesting and the results confirm that FQ affect tendon viability even at 8 days after discontinuation of treatment.
Fernandez-Cuadros et al has stated that pathlogy could be present since 2 hours of treatment to as far as 6 moths after stopping treatment (Fernández-Cuadros, M. E., Casique-Bocanegra, L. O., Albaladejo-Florín, M. J., Gómez-Dueñas, S., Ramos-Gonzalez, C., & Pérez-Moro, O. S. (2019). Bilateral Levofloxacin-Induced Achilles Tendon Rupture: An Uncommon Case Report and Review of the Literature. Clinical Medicine Insights: Arthritis and Musculoskeletal Disorders, 12, 1179544119835222). You could add this paragraph on introduction.
Material and methods are clearly explained, that I have nothing more to add.

Validity of the findings

The authors have stated that FQ inhibit tenocyte viability and inhibits synthesis of proteoglycans.
In discussion, Corrao et al 2006 was cited. You should describe that although you have seen that latency of recovery after FQ therapy is not observed at 8 days, other authors have described even FQ effect at 6 moths of FQ treatment (Fernandez-Ciuadros et al).

Additional comments

The manuscript is well written. The results are in accordance to the objectives of the study.
Discussion is linked to the matter of the manuscript.
Conclussion is not concise, some authors have been cited in such a section (Bank et al, Khaliq and Zhanel et al). I strongly suggest that the paragraph where authors are referenced should go in discussion part. I see techically innapropriate to cite/reference authors in conslussion section since this part is reserved to the final results of the manuscript, not for observations from other authors.
Some other paragraphs of conclusion refer to strength and limitations of the study. I would suggest authors to put those paragraphs in Discussion section, and to limit Conslussion to the main results observed in the study (viability of tenocytes at 96h and 8 days and PG synthesis) based on biochemical and gene evaluation.

Reviewer 3 ·

Basic reporting

1. Overall:
-This manuscript is fairly clearly written. A few specific comments:
-line 81: write out “approx.”
-At the start of a sentence, don’t use abbreviations or number (write-them out).
-Line 329: insert “in” between reduced and equine.
-Line 334: Please reword the first sentence of the conclusion for clarity

1. Introduction:
- very thorough.
2. Results:
-Effects on tenocyte viability (line 212): please specify that this is compared to controls.
-Line 231-232: is the change in versican mRNA in the treated or controls?
3. Discussion:
-The discussion is too long and includes information that is not relevant to this publication. For instance, on line 256, the authors make reference to ibuprofen and its affects on FQ accumulation. This is not relevant to this paper, nor is ibuprofen used in horses with any regularity.
-The mention of renal failure and renal transplant also seems misplaced in this paper.
-Removal of extraneous information will shorten this section and make it easier to read.
-Line 314: Since the study is cited, please provide a suggestion as to why the contradictory results between the current study and the Juras study.
4. Conclusion:
-extremely long and includes a lot of extraneous information. Would suggest making it more succinct rather than a repeat of what is found in the discussion.
5. Figures:
- clear and well described

Experimental design

1. The study begins to fill the gap in our knowledge regarding the potential toxicity of FQs on tendons in horses.
2. Suggest adding the reason for the selection of ciprofloxacin concentrations along with the aims in the introduction or in the methods section when the specific concentrations are first mentioned.
3. Methods are clear. One specific area that could be more clear is the “RNA extraction and reverse transcriptase” section. The authors state “in a separate of experiments.” Does this mean a completely different culture and were the treatments the same?
4. With respect to doses, are these really clinically relevant with respect to the concentrations that actually reach the tissue. The authors cite studies done in humans but no PK studies in horses. Furthermore, the concentrations that the authors suggest the tendon is exposed to in vivo seems very high. In looking at the references, there are only a couple that describe tendon concentrations and they are for other FQs and concentrations appear lower. In my mind this raises the question of the clinical applicability of these concentrations. Justification for doses should be more clear or at the very least acknowledgement that these may or may not be clinically relevant.

Validity of the findings

1. While this is a good first step in understanding the effects of FQs (specifically Ciprofloxacin) on tendon cells, the findings and the significance are a bit overstated.
-The findings from this study show an effect on tenocyte viability but the authors appear to over speculate as to the mechanism.
-Line 275: the authors suggest that suboptimal concentrations of decorin may be contributing to tenocyte viability. Unless I missed something, they looked at mRNA expression (not concentrations which suggest protein) and even then, there was not a significant difference in decorin expression.
-Line 282: the authors comment that the study provides insight into the mechanism underlying the latency between cessation of FQ therapy and development of tendon pathology. Please either remove or provide data to back up this statement from this study.

Additional comments

This paper attempts to begin to address an issue of importance in veterinary medicine but one that we don’t have much information about, especially in horses.

---

## Round 0.2 · accepted · Accept

Thank you for your modifications to your manuscript. I have conferred with the reviewers and deem your manuscript is acceptable for publication.

The Section Editor noted that the following caould be addressed during the production process:

The axes of figures 5-15 are ambiguous: legends state that the graphs are logarithmic, but in that case, the equidistant tick marks refer to 100, 10 0.1, 100.2 ,.... 101 =10, etc. but this is not labelled, and unwary readers might interpret the tick marks as "2", "3",4", etc. (or 0.9, 0.8, 0.7, etc. for the marks below the line, instead of 10^-0.1, 10^-0.2, etc.). This should be clarified, either by including a proper logarithmic scale or by explicitly labeling a few of the tick marks.